# Myeloid GSK3α Deficiency Reduces Lesional Inflammation and Neovascularization during Atherosclerotic Progression

**DOI:** 10.3390/ijms252010897

**Published:** 2024-10-10

**Authors:** Sarvatit Patel, Nisarg Shah, Brooke D’Mello, Anson Lee, Geoff H. Werstuck

**Affiliations:** 1The Thrombosis and Atherosclerosis Research Institute, Hamilton, ON L8L 2X2, Canada; sarvatit.patel@uhn.ca (S.P.); nisargshah228@gmail.com (N.S.); imansonlee@gmail.com (A.L.); 2Temerty Faculty of Medicine, University of Toronto, Toronto, ON M5S 1A1, Canada; 3Department of Medicine, McMaster University, Hamilton, ON L8N 3Z5, Canada

**Keywords:** atherosclerosis, glycogen synthase kinases, inflammation

## Abstract

The molecular mechanisms by which cardiovascular risk factors promote the development of atherosclerosis are poorly understood. We have recently shown that genetic ablation of myeloid glycogen synthase kinase (GSK)-3α attenuates atherosclerotic lesion development in low-density lipoprotein receptor-deficient (Ldlr^−/−^) mice. However, the precise contributions of GSK3α/β in atherogenesis are not known. The aim of this study is to investigate the effect of GSK3α and/or β deficiency on lesional inflammation and plaque vascularization. Five-week-old female Ldlr^−/−^ mice were fed a high-fat diet for 10 weeks to establish atherosclerotic lesions. Mice were harvested at 15 weeks of age and atherosclerotic lesions were characterized. The results indicate that, in addition to significantly reducing plaque volume, GSK3α-deficiency decreases inflammation, reduces vasa vasorum density at the aortic sinus, and reduces plasma c-reactive protein (CRP) levels. GSK3β-deficiency is associated with decreased plasma CRP levels but does not affect lesional inflammation or vascularization. These results suggest GSK3α may be an applicable target for the development of novel anti-atherogenic therapies.

## 1. Introduction

Cardiovascular disease (CVD) is the leading cause of death globally and a significant burden to healthcare systems [1]. Atherosclerosis is characterized by the buildup of plaque within arterial walls and plays a significant role in the pathogenesis of CVD [2,3]. As atherosclerosis progresses, the arteries narrow, reducing blood supply to vital organs and tissues and potentially causing ischemic events such as myocardial infarction (MI) and stroke [1,4].

Macrophages are a major component of atherosclerotic plaques and play a crucial role in the inflammatory response [5]. Monocytes migrate to the site of inflammation where they polarize into M1 or M2 macrophages, depending upon microenvironmental stimuli [6]. M1 macrophages are pro-inflammatory, driving inflammation through cytokine production, whereas M2 macrophages are anti-inflammatory and help maintain tissue homeostasis [7,8]. During atherosclerosis, macrophages ingest cellular debris and oxidized low-density lipoprotein (LDL) particles, becoming lipid-laden foam cells. Foam cell apoptosis and necroptosis lead to the formation of a dense necrotic core that destabilizes the plaque [1]. Unstable plaques can rupture, causing blood clots, potentially leading to fatal consequences such as MI [9].

Despite extensive research, the underlying mechanisms that link atherosclerotic risk factors to cardiovascular events remain poorly understood. Current therapies are generally focused upon reducing risk factors and include statins to lower LDL levels, anti-hypertensive medications to reduce blood pressure, and various drugs to reduce blood glucose levels in individuals with diabetes [10]. Our group has previously identified a role for endoplasmic reticulum (ER) stress-induced glycogen synthase kinase (GSK)-3 activity in the initiation and progression of atherosclerosis [11].

GSK-3 is a ubiquitously expressed serine/threonine kinase with two isoforms, GSK3α and GSK3β, which play a crucial role in many cell regulation pathways, including macrophage polarization, the inflammatory response, and cell viability [12]. Previous research has shown the GSK3α/β regulates the expression of pro-inflammatory cytokines such as interleukin (IL)-1β [1,13]. Additionally, our group has shown the myeloid-specific GSK3α-deficient low-density lipoprotein receptor-deficient (Ldlr^−/−^) mice show increased lesional M2 macrophage polarization and decreased atherosclerotic plaque volumes [11].

The aim of this study is to examine the impact of myeloid-specific GSK3α/β deficiency on pro-inflammatory and micro-vascularization pathways. We hypothesize that myeloid-specific GSK3α deficiency may reduce atherogenesis through the modulation of these pathways in Ldlr^−/−^ mice.

## 2. Results

### 2.1. Myeloid-Specific GSK3α Deficiency Reduces Inflammation and Attenuates Atherosclerosis

As previously observed, myeloid GSK3α deficiency attenuates the development of atherosclerosis at the aortic sinus in Ldr^−/−^ mice (Figure 1) [8]. Deficiency of GSK3β has no significant effect on atherosclerotic volume. To explore the effects of GSK3α/β deficiency on lesional inflammatory response, the expression of markers related to inflammation was examined in cross sections of the aortic sinus. Expressions of these markers were quantified in each of the experimental groups using immunofluorescent staining with antibodies against nuclear factor kappa-light-chain-enhancer of activated B cells (NFκB), Nod-like receptor family pyrin domain containing 3 (NLRP3), or IL-1ꞵ (Figure 2A–C).

LMαKO and LMαβKO mice had significantly decreased levels of NFκB, NLRP3, and IL-1ꞵ compared to Lαβfl/fl (controls) (Figure 2D–F). LMβKO mice showed no changes in NFκB or NLRP3 but had elevated IL-1ꞵ expression compared to the Lαβfl/fl controls (Figure 2D–F). These findings suggest that myeloid GSK3α may promote inflammation and thereby accelerate the growth of the atherosclerotic plaque.

To determine the effects of GSK3α/β deficiency on systemic inflammatory responses, the plasma level of C-reactive protein (CRP) was quantified. LMαKO, LMβKO, and LMαβKO mice showed decreased plasma CRP levels compared to Lαβfl/fl controls (Figure 3). This finding suggests that both myeloid GSK3α and GSK3β may play a role in systemic inflammation.

### 2.2. Myeloid-Specific GSK3α Deficiency Reduces Vasa Vasorum Density

Cross sections of the aortic sinus were immunostained with an antibody against hypoxia-inducible factor (HIF)-1α (Figure 4A). LMαKO and LMαβKO mice had decreased levels of HIF-1α compared to Lαβfl/fl controls (Figure 4B). LMβKO mice showed no changes in HIF-1α compared to the Lαβfl/fl controls (Figure 4B). This suggests that myeloid GSK3α plays a role in activation of lesional HIF-1α which may affect atherogenesis.

To measure the vasa vasorum density at the aortic sinus and ascending aorta, aortic cross sections were immunostained with an antibody against endothelial marker, Von Willebrand factor (vWF), and the number of positively stained microvessels per cross section were quantified (Figure 5A). LMαKO and LMαβKO mice had relatively fewer positively stained microvessels compared to Lαβfl/fl controls (Figure 5B). LMβKO mice showed no change in microvessel numbers compared to the Lαβfl/fl controls. These findings suggest that myeloid GSK3α deficiency significantly reduced microvessel density (Figure 5B).

To further investigate the effect on vasa vasorum density, aortic cross sections were immunostained with an antibody against vascular endothelial growth factor (VEGF) (Figure 6A). LMαKO and LMαβKO mice had decreased levels of VEGF compared to Lαβfl/fl controls (Figure 6B). LMβKO mice showed no changes in VEGF levels compared to the Lαβfl/fl controls (Figure 6B). Together, these data suggest a primary role for GSK3α in the modulation of the HIF-1α-VEGF pathway that regulates vasa vasorum density.

## 3. Discussion

The results of this study suggest that myeloid-specific GSK3α (and GSK3αβ) deficiency reduces atherosclerosis and inflammation in atherosclerotic lesions and reduces vasa vasorum density at the aortic sinus in Ldlr^−/−^ mice. In contrast, GSK3β deficiency appears to have no significant effect on these variables. This suggests that deletion of GSK3α is the driving factor behind the observed phenotype in GSK3αβ double knockout mice.

Previous studies have shown that a buildup of unesterified cholesterol causes endoplasmic reticulum stress in macrophages, leading to the activation of GSK3α/β via the protein kinase R-like ER kinase (PERK) pathway [11]. GSK3 subsequently activates the pro-inflammatory response via the NFκB signaling cascade [14,15,16]. The results presented here suggest a correlation between inflammation and the stabilization/accumulation of HIF-1α and the upregulation of VEGF [17,18]. VEGF increases endothelial permeability and contributes to monocyte adhesion and also activates angiogenesis and neovascularization, crucial for atherosclerotic progression [19,20]. Our results elucidate GSK3α/β’s role in these pathways.

We observed that myeloid-specific GSK3α- and GSK3αβ-deficient Ldlr^−/−^ mice show lower expression of lesional NFκB, NLRP3, and IL-1β compared to controls, correlating with reduced lesion area during atherosclerosis. The downregulation of NFκB leads to decreased NLRP3 expression and reduced IL-1β secretion at injury sites [14]. IL-1β recruits monocytes and induces monocyte adhesion molecules such as vascular cell adhesion molecule (VCAM)-1 and P-selectin, thereby promoting inflammation [21]. Consequently, GSK3α deletion reduces monocyte recruitment/adhesion and inflammation, aligning with previous research indicating that GSK3α deficiency promotes an M2 macrophage phenotype [12].

Furthermore, by visualizing vWF, we observed that myeloid-specific GSK3α- and GSK3αβ-deficient Ldlr^−/−^ mice exhibit reduced microvascularization at the aortic sinus. GSK3α and GSK3αβ deficiency lowers VEGF expression in Ldlr^−/−^ mice. Previous research has primarily focused on GSK3β’s role in VEGF expression and angiogenesis in endothelial cells [22,23]. However, our data suggest that GSK3α may play a more significant role in VEGF expression and angiogenesis in atherogensis than previously recognized.

Our findings suggest that GSK3α may be a suitable target for developing anti-atherosclerotic therapies. Specifically targeting GSK3α may be advantageous as GSK3β is known to regulate important functions, including roles in the Wnt signaling pathway, glycogen metabolism regulation, and apoptosis [24]. Previous studies have shown that whole-body GSK3β knockout mice die during gestation, underscoring the necessity of preserving GSK3β’s functions [12]. A therapeutic focus on GSK3α may therefore limit unwanted side effects.

In summary, these findings are consistent with previous studies identifying atherosclerosis as a chronic inflammatory immune-mediated disease [2,3,4,25]. These results suggest that myeloid-specific GSK3α deficiency reduces inflammation in lesions and vasa vasorum density at the aortic sinus in Ldlr^−/−^ mice. Future research should aim to quantify HIF-1α expression in GSK3α/β-deficient Ldlr^−/−^ mice to better understand the reduction in VEGF expression observed with GSK3α and GSK3αβ deficiency. Additionally, mechanistic studies to determine how GSK3α promotes M1 macrophage polarization are warranted. Overall, these insights will advance our understanding of GSK3 isoforms, whilst opening new avenues for therapeutic intervention in atherosclerosis.

## 4. Materials and Methods

### 4.1. Mouse Models

Myeloid-specific GSK3α- and/or GSK3β-deficient mice, were created by our lab [26]. This includes Ldlr^−/−^ mice with loxP-flanked GSK3α gene (Ldlr^−/−^GSK3α^fl/fl^) crossed with mice expressing a single copy of the Cre recombinase gene controlled by the myeloid-specific lysozyme M promoter (Ldlr^−/−^LyzMCre^+/−^GSK3α^fl/fl^). By using this breeding method, we were able to generate the Ldlr^−/−^ myeloid-specific GSK3α knockout mice (Ldlr^−/−^LyzMCre^+/−^GSK3α^fl/fl^ or LMαKO). The Ldlr^−/−^ myeloid-specific GSK3β-deficient mice were bred similarly to obtain Ldlr^−/−^ myeloid-specific GSK3β knockout mice (Ldlr^−/−^LyzMCre^+/−^GSK3β^fl/fl^ or LMβKO). Breeding strategies using the above mice were also used to generate Ldlr^−/−^ myeloid-specific GSK3α/β-deficient mice (Ldlr^−/−^LyzMCre^+/−^GSK3α^fl/fl^GSK3β^fl/fl^ or LMαβKO) and the control Ldlr^−/−^ GSK3α/β floxed mice (Ldlr^−/−^GSK3α^fl/fl^GSK3β^fl/fl^ or Lαβfl/fl). All the mouse strains described above exist in a C57Bl/6 genetic background. All animal experiments were pre-approved by the McMaster University Animal Research Ethics Board. All experiments conform with the guidelines and regulation of the Canadian Council on Animal Care.

### 4.2. Atherosclerotic Progression Model

Five-week-old female mice (Lαβfl/fl, LMαKO, LMβKO, and LMαβKO) were fed a high-fat diet (HFD) containing 21% fat and 0.2% cholesterol, with 42% calories from fat (TekLad TD97363, Inotiv, Madison, WI, USA) for 10 weeks to establish atherosclerotic plaques. All mice were granted access to water ad libitum. Eight mice per experimental group were harvested at 15 weeks of age. Mice were fasted for 6 h prior to sacrifice. Body weight was measured, and 3% isoflurane was used to anesthetized mice. Blood was collected via cardiac puncture and livers and perigonadal fat pads were harvested and weighed (Appendix A). The vasculature was flushed with 1x phosphate-buffered saline (PBS) and perfusion fixed with 10% neutral buffer formalin. Hearts and aortas, along with other tissues, were collected and fixed in formalin.

### 4.3. Characterization of Aortic Lesions

Hearts and aortas from eight mice per experimental group were embedded in paraffin and 5 μm sections were collected onto slides, starting from the aortic sinus and moving up the ascending aorta [27]. For immunofluorescent staining, sections were deparaffinized and antigen retrieval was performed using antigen-unmasking solution (Vector laboratories H-3300, Burlington, ON, Canada). Sections were blocked in 10% goat serum and then immunostained overnight with primary antibodies against the pro-inflammatory markers NF-κB diluted 1:50 (Santa Cruz Biotechnology sc-8008, Dallas, TX, USA), inflammasome marker NLRP3 diluted 1:100 (Invitrogen MA5-32255, Mississauga, ON, Canada), and IL-1β diluted 1:100 (Invitrogen P420B, Mississauga, ON, Canada). To investigate the effect on vasa vasorum density, sections were immunostained overnight with primary antibodies against HIF-1α (NOVUSNB100-105, Toronto, ON, Canada), vWF (Aligent Technologies GA52761-2, Mississauga, ON, Canada) and VEGF (Santa Cruz Biotechnology sc-7269, Dallas, TX, USA). Sections were then incubated with secondary antibodies Alexa Fluor 488 goat anti-mouse IgG diluted 1:250 (Thermofisher Scientific A11001, Mississauga, ON, Canada), Alexa Fluor 488 goat anti-rabbit IgG diluted 1:250 (Thermofisher Scientific A11008, Mississauga, ON, Canada) for 2 h, and then stained with the DAPI (diluted 1:5000) (Invitrogen D1306, Mississauga, ON, Canada). Slides were mounted with Fluoromount Aqueous Mounting Medium (Sigma F4680, Oakville, ON, Canada) and stored at 4 °C in the dark. Separate slides were stained with pre-immune IgG instead of primary antibodies to control for non-specific staining. Images of the stained sections were collected using the Leica STELLARIS 5 confocal microscope. All images were taken at 100× magnification (10× objective, 10× eyepiece). Image J 1.52q software was used to quantify immunofluorescent staining. Four sections were analyzed per mouse and three images were captured per section (one for each lesion/leaflet). The threshold for image analysis was based on the quality of antibody staining in comparison to negative controls, and this threshold was consistently applied across all samples. The total area of the plaque and the total stained area within that plaque were determined, and then the percentage of stained area for each section was calculated. The percentage of stained area was averaged across the four sections for each animal to represent the data for each sample.

### 4.4. Determination of Plasma CRP

The plasma CRP levels were determined by using the Mouse C-Reactive Protein (CRP) ELISA Kit (Crystal Chem 80634, Elk Grove Village, IL, USA). Assays were performed (n = 8/experimental group) according to manufacturer’s instructions.

### 4.5. Statistical Analysis

All statistical analysis was performed in GraphPad Prism software (version 9.3.1). All data was analyzed by a one-way ANOVA, followed by the Tukey’s multiple comparison test between all groups. All error bars on graphs represent the standard error of the mean (SEM). For all experiments, a *p* value lower than 0.05 was considered statistically significant. * *p* < 0.05, ** *p* < 0.01, *** *p* < 0.001, **** *p* < 0.0001.

## Figures and Tables

**Figure 1 ijms-25-10897-f001:**
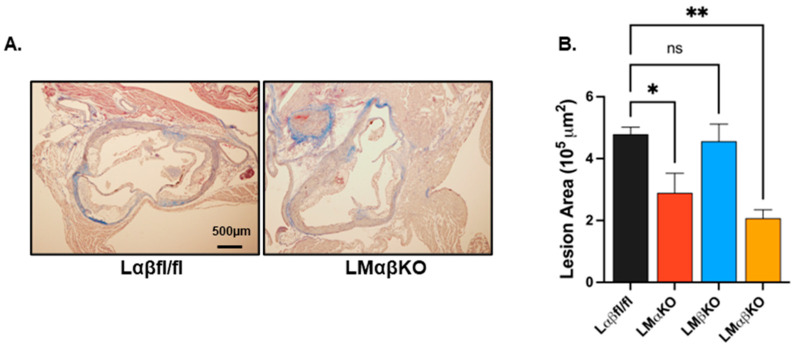
Myeloid-specific glycogen synthase kinase (GSK)-3α deficiency reduces lesion area at the aortic sinus in low density lipoprotein receptor knockout (Ldlr^−/−^) mice. (**A**) Representative Mason’s trichrome-stained sections of aortic sinus from GSK3αβ floxed (Lαβfl/fl) and GSK3αβ double knockout (LMαβKO) mice. Scale bar = 500 µm. (**B**) Quantification of lesion area at the aortic sinus of Ldlr^−/−^ control mice (Lαβfl/fl), or myeloid GSK3α (LMαKO), β (LMβKO), or α and β (LMαβKO) was genetically deleted. n = 6–8/experimental group, * *p* < 0.05, ** *p* < 0.01, ns—not significant (One-way ANOVA).

**Figure 2 ijms-25-10897-f002:**
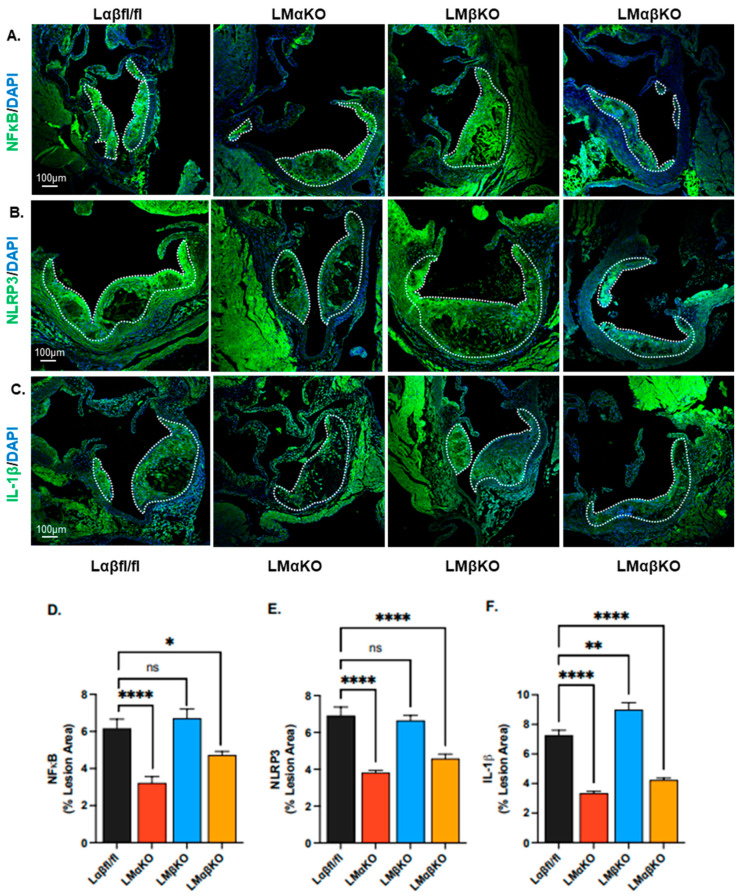
Myeloid-specific GSK3α deficiency reduces lesional inflammation. Representative sections of aortic sinus immunostained with an antibody against (**A**) NFκB (green), (**B**) NLRP3 (green) or (**C**) IL-1β (green). Lesional areas are indicated by a dotted white line. Scale bar = 100 µm. Quantification of (**D**) NFκB and (**E**) NLRP3 and (**F**) IL-1β-stained area normalized to the total lesion area. n = 8/experimental group; mean ± SEM; * *p* < 0.05, ** *p* < 0.01, **** *p* < 0.0001, ns—not significant (One-way ANOVA).

**Figure 3 ijms-25-10897-f003:**
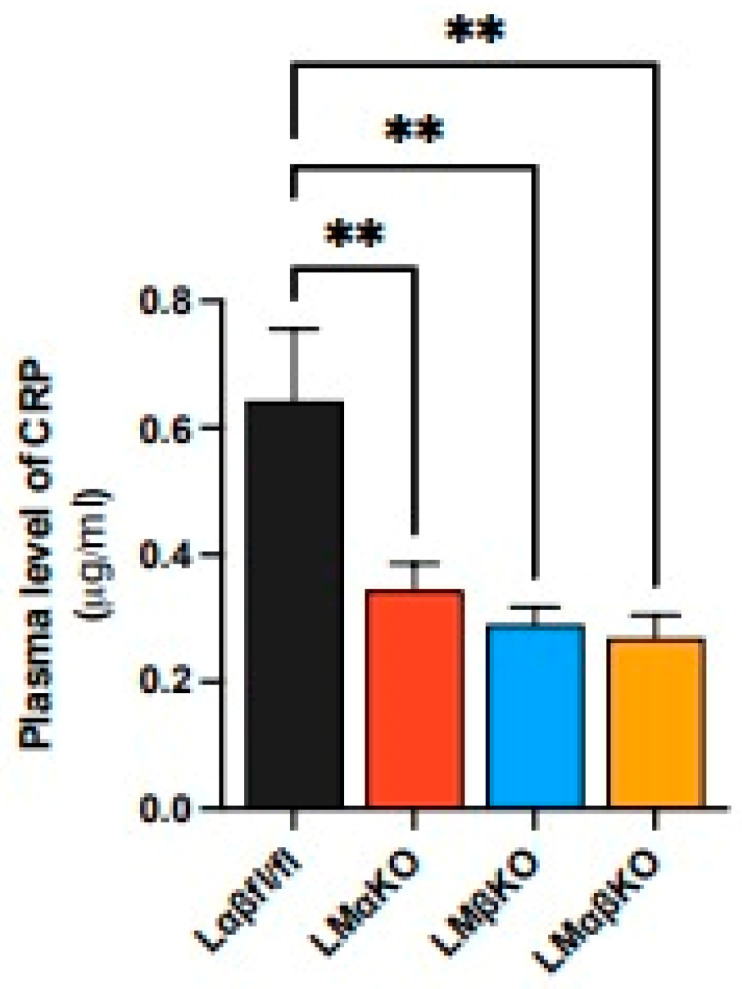
Myeloid GSK3α and GSK3β both modulate CRP expression. Quantification of plasma levels of C-reactive protein (CRP). n = 8/experimental group; mean ± SEM; ** *p* < 0.01 (One-way ANOVA).

**Figure 4 ijms-25-10897-f004:**
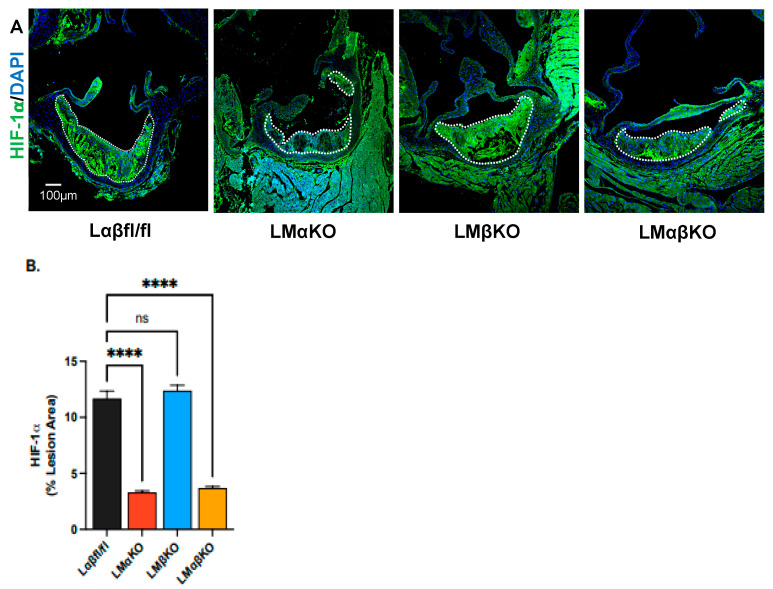
Myeloid-specific GSK3α deficiency is associated with reduced HIF1α in atherosclerotic lesions. Representative sections of aortic sinus immunostained with an antibody against (**A**) HIF-1α (green). Lesional areas are indicated by a dotted white line. Scale bar = 100 µm. Quantification of (**B**) HIF-1α-stained area normalized to the total lesion area. n = 8/experimental group; mean ± SEM; **** *p* < 0.0001, ns—not significant (One-way ANOVA).

**Figure 5 ijms-25-10897-f005:**
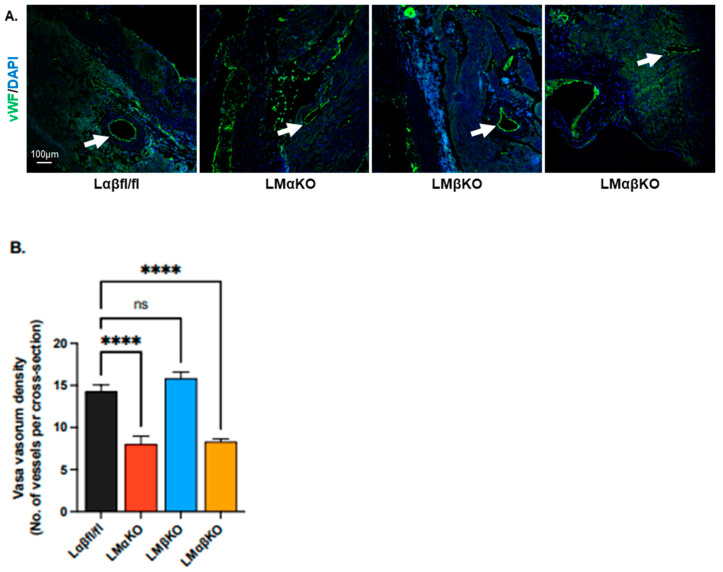
Myeloid-specific GSK3α deficiency reduces micro-vascularization at the aortic sinus. Representative sections of aortic sinus immunostained with an antibody against (**A**) vWF (green). Microvessels are indicated by arrows. Scale bar = 100 µm. Quantification of (**B**) vasa vasorum density (number of vessels per cross section). n = 8/experimental group; mean ± SEM; **** *p* < 0.0001, ns—not significant (One-way ANOVA).

**Figure 6 ijms-25-10897-f006:**
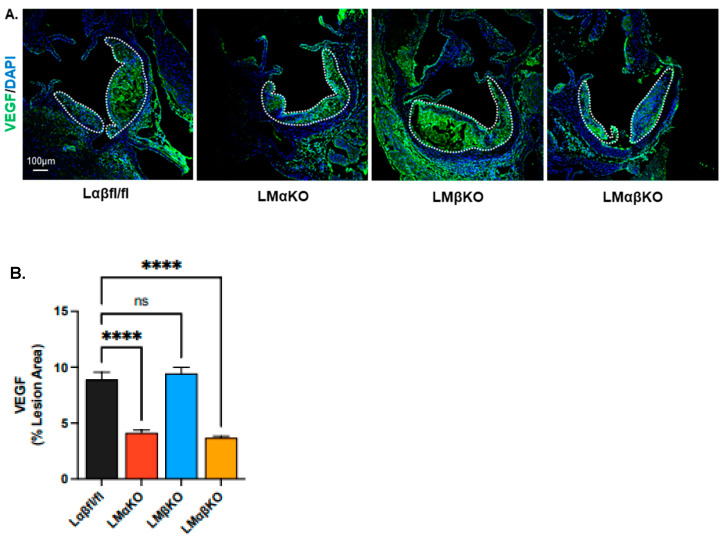
Myeloid-specific GSK3α deficiency reduces VEGF expression at the aortic sinus. Representative sections of aortic sinus immunostained with an antibody against (**A**) VEGF (green). Lesional areas are indicated by a dotted white line. Scale bar = 100 µm. Quantification of (**B**) VEGF-stained area is normalized to the total lesion area. n = 7/experimental group; mean ± SEM; **** *p* < 0.0001, ns—not significant (One-way ANOVA).

## Data Availability

Data are contained within the article.

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
