# Peer review of "Myeloid GSK3α Deficiency Reduces Lesional Inflammation and Neovascularization during Atherosclerotic Progression"

_ijms, 2024, doi:10.3390/ijms252010897_

Round 1

Reviewer 1 Report

Comments and Suggestions for Authors

Review of the manuscript entitled: Myeloid GSK3α-Deficiency Reduces Lesional Inflammation and Neovascularization During Atherosclerotic Progression. The manuscript touches on a important problem. In my opinion, the manuscript is interesting but some corrections are necessary.

1.      It is crucial to state the purpose of the research. Please add the exact aim of the manuscript in the end of introduction and in abstract. This will improve the clarity of why the work was done. e.g. "The aim of the present study was to ...". But generally the introduction and abstract are written correctly and very clearly.

2.      Figures 2, 4, 5, 6 add magnification and scale bar to microphotographs. On this page you will find information how to calculate the magnification of a microscope https://carson.com/optics-university/microscope-hub/how-to-calculate-the-magnification-of-a-microscope/

3.      All abbreviations should be explained when they are first used e.g. line 111 and 156 -  HIF-1α. Check the entire manuscript and other abbreviations.

4.      I strongly suggest adding more objective markers such as qPCR or western blot. Images are very subjective, and easy to cheat. In its current form, the manuscript can be published as a short communication but not as a full article. Consider changing the article type, or if it is possible add objective molecular methods (qPCR or WB) and then can be published as full article.

5.      Please attenuate the discussion because it is speculative. In line 146 maybe not “demonstrates” or better “suggests”. Please correct the whole discussion.

6.      In materials and methods, please add the catalog numbers of key reagents, especially antibodies.

Author Response

Response to Reviewer 1

The authors would like to thank Reviewer 1 for their constructive and helpful comments. We have addressed each of the comments as indicated in the point-by-point response presented below. For completeness, the reviews are pointed in their entirety (in italics). Responses, where appropriate, are indicated by the symbol “â–º”. Major additions to the manuscript are indicated in RED in the revised text.

Reviewer 1

Review of the manuscript entitled: Myeloid GSK3α-Deficiency Reduces Lesional Inflammation and Neovascularization During Atherosclerotic Progression. The manuscript touches on a important problem. In my opinion, the manuscript is interesting but some corrections are necessary.

  1. It is crucial to state the purpose of the research. Please add the exact aim of the manuscript in the end of introduction and in abstract. This will improve the clarity of why the work was done. e.g. "The aim of the present study was to ...". But generally the introduction and abstract are written correctly and very clearly.

â–ºThe aims of this study are now more clearly stated in the abstract and the revised Introduction.

  1. Figures 2, 4, 5, 6 add magnification and scale bar to microphotographs. On this page you will find information how to calculate the magnification of a microscope https://carson.com/optics-university/microscope-hub/how-to-calculate-the-magnification-of-a-microscope/

â–ºWe have included a scale bar on the control image in each set of figures. This scale bar is relevant to all of the images in each set. All images were taken at 100x magnification (10x objective, 10x eyepiece). This information is included in the revised Materials and Methods section.

  1. All abbreviations should be explained when they are first used e.g. line 111 and 156 -  HIF-1α. Check the entire manuscript and other abbreviations.

â–ºAll abbreviations are now defined upon first used in the manuscript.

  1. I strongly suggest adding more objective markers such as qPCR or western blot. Images are very subjective, and easy to cheat. In its current form, the manuscript can be published as a short communication but not as a full article. Consider changing the article type, or if it is possible add objective molecular methods (qPCR or WB) and then can be published as full article.

â–ºWe agree that the inclusion of “more objective” techniques – qPCR and/or immunoblot – would strengthen the manuscript. However, application of these techniques to the characterization of murine atherosclerotic lesions is technically difficult (in the case of qPCR – requiring laser capture techniques) or not feasible (in the case of immunoblotting – die to the very limited amount of protein that could be captured from lesional cells). Furthermore, such studies would require a new set of animal experiments to isolate the required material. Future studies, will be focused upon a more cell specific, and more robust characterization of the roles of GSK3αβ.

  1. Please attenuate the discussion because it is speculative. In line 146 maybe not “demonstrates” or better “suggests”. Please correct the whole discussion.

â–ºAs requested, the Discussion has been rewritten to “tone down” the interpretation of these findings.

  1. In materials and methods, please add the catalog numbers of key reagents, especially antibodies.

â–ºCatalog numbers have been added to all key reagents including antibodies.

Thank you again for taking the time to review this manuscript.

Reviewer 2 Report

Comments and Suggestions for Authors

The paper is interesting and well written. The authors investigated investigated the effect of GSK3α and/or β deficiency on lesional inflammation and plaque vascularization. . The results confirmed that GSK3α-deficiency decreases inflammation and reduces vasa vasorum density at the aortic sinus, suggesting that GSK3α may be a potential target for a novel anti-atherogenic therapies. The study iw well structured, the methodology is adequate and coerent with the endpoints. References are adequate. The figures are well defined. I suggest to discuss the role of Th17 cells in chronic inflammatory immune-mediated diseases and plaque is a chronic inflammatory immune-mediated process (see and add as referecne paper by Murdaca et al concerning Th17 cells in chronic inflammatory immune-mediated diseases). Finally, I sugges to discuss the role of TNF alpha inhibitor in controlling lipid levels and in improving plaque (see and add as references papers by Murdaca et al concerning the efficacy and safety of TTNF alpha inhibitors) in order to confirm the importance of target therapy.

Comments on the Quality of English Language

Minor english editing

Author Response

Response to Reviewer 2

The authors would like to thank Reviewer 2 for their constructive and helpful comments. We have addressed each of the comments as indicated in the point-by-point response presented below. For completeness, the reviews are pointed in their entirety (in italics). Responses, where appropriate, are indicated by the symbol “â–º”. Major additions to the manuscript are indicated in RED in the revised text.

Reviewer 2

The paper is interesting and well written. The authors investigated the effect of GSK3α and/or β deficiency on lesional inflammation and plaque vascularization. . The results confirmed that GSK3α-deficiency decreases inflammation and reduces vasa vasorum density at the aortic sinus, suggesting that GSK3α may be a potential target for a novel anti-atherogenic therapies. The study is well structured, the methodology is adequate and coherent with the endpoints. References are adequate. The figures are well defined. I suggest to discuss the role of Th17 cells in chronic inflammatory immune-mediated diseases and plaque is a chronic inflammatory immune-mediated process (see and add as reference paper by Murdaca et al concerning Th17 cells in chronic inflammatory immune-mediated diseases). Finally, I suggest to discuss the role of TNF alpha inhibitor in controlling lipid levels and in improving plaque (see and add as references papers by Murdaca et al concerning the efficacy and safety of TTNF alpha inhibitors) in order to confirm the importance of target therapy

â–ºWe have included a sentence, “….these findings are consistent with previous studies identifying atherosclerosis as a chronic inflammatory immune mediate disease” and have refenced Murdaca et al. (2011) and several other relevant papers. The authors feel that delving into the efficacy and safety of TNFα inhibitors, while interesting, is beyond the scope of this particular study.

Thank you again for taking the time to review this manuscript.

Reviewer 3 Report

Comments and Suggestions for Authors

The manuscript focused on the role of GSK3a/b in atherosclerotic lesion development. Using an in vivo mouse model (ldlr-/-) on high fat diets, crossed with GSK3a/b depletion. Primary endpoints were lesion properties, serum cytokines, and vascular parameters. The methods section is appropriately detailed and statistics largely appropriate. The results are well described and the discussion appropriate. However, several key pieces of information were unclear. These and other reviewer comments are detailed below:

1) Replicate numbers are not well described. They are needed for all experiments.

2) Figure 1 should specify the statistical test used, similar to the other Figures.

3) A brief figure showing representative lesions used for volume measurements would be useful in Figure 1.

4) Much greater detail is needed for methods regarding immunofluorescent quantification of % Lesion Area. How many images/sections per animal, how were thresholds determined, were they consistent, were they automated or manual, if manual were the researchers masked to genotype, etc. The specific analysis methods should be described as well, mentioning “ImageJ was used” is not sufficient.

5) GSK3a plasma CRP results could be mentioned in Abstract

6) “To examine the levels of hypoxia in the vascular wall, cross sections of the aortic sinus were immunostained with an antibody against HIF-1α (Figure 4A).” HIF is regulated by many things, while it is possible changes are related to hypoxia, that can’t be assumed.

7) The immunofluorescent images are a little dim, it would help to brighten them (linearly and consistently)

8) “The results presented here suggest that this inflammation results in hypoxia,” this statement is too strong. It is consistent with the results.

Comments on the Quality of English Language

9) There are a few typos throughout, although the manuscript is overall quite well written.

Author Response

Response to Reviewer 3

The authors would like to thank Reviewer 3 for their constructive and helpful comments. We have addressed each of the comments as indicated in the point-by-point response presented below. For completeness, the reviews are pointed in their entirety (in italics). Responses, where appropriate, are indicated by the symbol “â–º”. Major additions to the manuscript are indicated in RED in the revised text.

Reviewer 2

The manuscript focused on the role of GSK3a/b in atherosclerotic lesion development. Using an in vivo mouse model (ldlr-/-) on high fat diets, crossed with GSK3a/b depletion. Primary endpoints were lesion properties, serum cytokines, and vascular parameters. The methods section is appropriately detailed and statistics largely appropriate. The results are well described and the discussion appropriate. However, several key pieces of information were unclear. These and other reviewer comments are detailed below:

1) Replicate numbers are not well described. They are needed for all experiments.

â–ºReplicate numbers for each experiment are now more clearly indicated in the Materials and Methods section and in individual figure legends.

2) Figure 1 should specify the statistical test used, similar to the other Figures.

â–ºThe statistical test used in Figure 1 was “one way ANOVA”. This is now indicated in the revised figure legend.

3) A brief figure showing representative lesions used for volume measurements would be useful in Figure 1.

â–ºMason’s trichrome stained images form representative control (Lαβfl.fl) and double GSK3αβ KO (LMαβKO) mice have been included in revised Figure 1.

4) Much greater detail is needed for methods regarding immunofluorescent quantification of % Lesion Area. How many images/sections per animal, how were thresholds determined, were they consistent, were they automated or manual, if manual were the researchers masked to genotype, etc. The specific analysis methods should be described as well, mentioning “ImageJ was used” is not sufficient.

â–ºthe following paragraph has been added to the Materials and Methods section:

“Four sections were analyzed per mouse and three images were captured per section (one for each lesion/leaflet). The threshold for image analysis was based on the quality of antibody staining in comparison to negative controls, and this threshold was consistently applied across all samples. Total area of the plaque and the total stained area within that plaque were determined and then the percentage of stained area for each section was calculated. The percentage of stained area was averaged across the four sections for each animal to represent the data for each sample.”

5) GSK3a plasma CRP results could be mentioned in Abstract

â–ºThe CRP results, as they pertain to GSK3α KO, are mentioned in the revised abstract.

6) “To examine the levels of hypoxia in the vascular wall, cross sections of the aortic sinus were immunostained with an antibody against HIF-1α (Figure 4A).” HIF is regulated by many things, while it is possible changes are related to hypoxia, that can’t be assumed.\

â–ºReviewer 3 is correct. The phrase “to examine the levels of hypoxia in the vascular wall…” has been deleted from this sentence.

7) The immunofluorescent images are a little dim, it would help to brighten them (linearly and consistently)

â–ºAll of the figures have been revised. The images have been consistently enhanced and enlarged to increased clarity. In addition, lesion areas have been outlined to improve clarity.

8) “The results presented here suggest that this inflammation results in hypoxia,” this statement is too strong. It is consistent with the results.

â–ºThis sentence has been modified to “The results presented here suggest a correlation between inflammation and the stabilization/accumulation of HIF-1α and the upregulation of VEGF”.

Thank you again for taking the time to review this manuscript.

Round 2

Reviewer 1 Report

Comments and Suggestions for Authors

the authors improved the manuscript but in the future remember about more objective methods ;)

Reviewer 3 Report

Comments and Suggestions for Authors

Authors have addressed concerns.